# Exploring galactagogue use among breastfeeding women: Insights from an observational study

**Agnieszka Garbacz**[1], **Paweł Juszczak**[1], **Marcin Nowicki**[2]*, **Przemysław Łukasz Kowalczewski**[3], **Magdalena Człapka-Matyasik**[1]

**1** Department of Human Nutrition and Dietetics, Poznań University of Life Sciences, Poznań, Poland,
**2** Institute of Agriculture, University of Tennessee, Knoxville, TN, United States of America, **3** Department of Food Technology of Plant Origin, Poznań University of Life Sciences, Poznań, Poland

* magdalena.matyasik@up.poznan.pl, mnowicki@utk.edu

**Data Availability Statement:** The summary of anonymous data is included within the report. Raw anonymous data may be requested from the Corresponding Authors upon reasonable request,

## Abstract

Breastfeeding is the optimal form of infant nutrition and remains a critical topic of discussion. In the case of lactation problems, women can be assisted by plant galactagogues designed to induce, sustain, and increase lactation. Reports on the popularity, awareness and knowledge of galactagogues are limited. Therefore, this study aimed to analyze the use of galactagogues in the diet among breastfeeding women. The online survey was conducted using anonymized questionnaires, with results collected in the Spring of 2021. Fifty-two women aged 20 to 50 who fed naturally participated in the study, with 69% of respondents being familiar with galactagogues and 42% having used them. The most often indicated herbs were fennel (68%) and anise (45%). Galactagogues were used as ready-to-use herbal lactation mixes (73%). Women found them effective in stimulating lactation (82%) and purchased them in pharmacies (64%) or herbal stores (36%). Respondents were encouraged to use galactagogues by media (32%) and family and friends (45%). Women reported that greater knowledge (57%) would be essential to effectively encourage galactagogues. Breastfeeding women were positive about using plant-based galactagogues and considered them effective. A critical barrier identified by those not using galactagogues was their need for knowledge. The information campaign for pregnant women should include galactagogues as lactation-stimulating herbs.

## 1. Introduction

Breastfeeding as the optimal form of infant nutrition is a topic whose relevance continues to grow. As per the recommendations of the WHO or the Polish Society of Gastroenterology, Hepatology, and Nutrition on exclusive breastfeeding up to the sixth month of a child's life [1, 2], its promotion and support are essential. Thus, an important role is uncovered for galactagogues, the plant lactogenic compounds. About 15% of American women used galactagogues, whereas in Norway -the country with the highest breastfeeding rate globally- as much as 43% [3, 4]. Stimulation and maintenance were recognized as the most critical problems with

due to involvement of human respondents. Data requests may also be submitted to (non-author data contact) prof. Magdalena Zielińska-Dawidziak (magdalena.zielinska-dawidziak@up.poznan.pl), Chairperson of the Food and Nutrition Technology Discipline Committee at Department of Food Biochemistry and Analysis, Faculty of Food Science and Nutrition, Poznań University of Life Sciences, Poznań, Poland.

**Funding:** This study was financially supported by the University of Tennessee [https://www.utk.edu] in the form of an Open Access Publication Fund award received by MN. No additional external funding was received for this study.

**Competing interests:** The authors have declared that no competing interests exist.

lactation. Therefore, breastfeeding women seek the support of various galactagogues, which, as lactation stimulants, help induce, sustain, and increase it.

The Internet remains the primary source of information on plant galactagogues [5]. Nevertheless, it was a source of as much valuable information as the contradictory one. The exact physiological mechanism of action of galactagogues is yet to be fully explained. However, several studies are available in which the authors have attempted to evaluate the effectiveness and effect of galactagogues on the stimulation and maintenance of lactation. In Anglo-Saxon countries, fenugreek was the most commonly used galactagogue [6]. It was also used in Indonesia, where green beans and brown sugar drinks were equally popular [6]. In Poland, fennel, cumin, and aniseed were the most popular. Fenugreek and rutabaga are also often served [6]. The effectiveness of herbal teas containing lemon balm, nettle, fennel, caraway, or aniseed extracts was analyzed. Their use helped increase lactation without adverse side effects [7, 8]. However, galactagogues used during lactation sometimes result in diarrhoea, headache, dry mouth, or stomach cramps [5, 9, 10]. A commonly used galactagogue is the previously mentioned fennel. Its action, however, may not always be advisable, as fennel oil, through the trans-anethole it contains, may exhibit intense abortifacient activity [11]. Aniseed, due to its anethole content, may exert toxic effects on the nervous and gastrointestinal systems [12]. More studies are still needed to confirm the efficacy of women's use of galactagogues and to investigate galactagogues' safety. Polish data regarding the popularity and use of plant-based galactagogues is also needed.

Given the multiplicity and diversity of available information, galactagogues' knowledge and dietary use among lactating women were analyzed. This study aimed to determine respondents' attitudes towards galactagogues use and opinions about their effectiveness.

## 2. Materials and methods

### 2.1 The study sample

The survey was conducted among 52 women between the ages of 20 and 50 who had breastfed naturally in the past or at the time of the survey (Table 1). All procedures followed the ethical standards of the institutional and national research committees and the Helsinki Declaration. This study was not a medical experiment, so it was exempt from ethical approval from the Poznan University of the Medical Sciences Bioethics Committee according to Polish laws and GCP regulations (decision number: 261 527/20). The participants consented to participate in the study with a digitally informed consent form. Such informed consent was obtained from all the respondents involved in the study. Data were collected between February 1st and April 30th, 2021, using an anonymous questionnaire created in Google Forms and distributed online. Google Forms fulfil the condition of anonymity; they ensure that the surveys created with them are entirely anonymous ("Collect email addresses" is not checked). Therefore, the authors could not identify individual participants during or after data collection.

As the survey took place during the COVID-19 outbreak, a safe and effective questionnaire was distributed through social media platforms, forums, and discussion groups. Recruitment was also conducted using the snowball sampling method described previously, where the new subjects were recruited by others to form a part of the sample [13]. The questionnaire was disseminated to Polish-speaking nursing mothers and pregnant women. In all mentioned places, a reminder and a request to complete it were posted with a frequency of once a week from February to April 2021.

For the statistical analysis, respondents were divided into those using galactagogues (GAG), those not using galactagogues (nGAG), and those familiar with galactagogues' use (fGAG).

**Table 1. Sociodemographic characteristics of the respondents.**

| Variable | Number of total participants N = 52 (%N) | | | | |
|---|---|---|---|---|---|
| | Study sample | Users (GAG) | Non-users (nGAG) | Women with knowledge of galactagogues (fGAG) | |
| | n (%) | n (%) | n (%) | n (%) | p¹ |
| n (%) | n = 52 (100%) | n = 22 (42%) | n = 30 (58%) | n = 36 (69%) | |
| Education level: | | | | | $X^2$ (3, N = 52) = 4.5, p = .21 |
| Primary | 1 (2) | 1 (5) | 0 (0) | 0 (0) | |
| Vocational | 3 (6) | 2 (9) | 1 (3) | 2 (6) | |
| Secondary | 19 (37) | 5 (23) | 14 (47) | 10 (28) | |
| Higher | 29 (56) | 14 (64) | 15 (50) | 23 (64) | |
| Place of residence: | | | | | $X^2$ (3, N = 52) = 3.8, p = .29 |
| City >100,000 residents | 23 (44) | 13 (59) | 10 (33) | 19 (53) | |
| City 20–100,000 residents | 13 (25) | 4 (18%) | 9 (30) | 7 (19) | |
| City <20,000 residents | 5 (10)table | 1 (5) | 4 (13) | 2 (6) | |
| Countryside | 11 (21) | 4 (18) | 7 (23) | 8 (22) | |
| Age: | | | | | $X^2$ (2, N = 52) = 2.9, p = .24 |
| 20 through 29 years old | 17 (33) | 10 (45) | 7 (23) | 15 (42) | |
| 30 through 39 years old | 27 (52) | 9 (41) | 18 (60) | 16 (44) | |
| 40 through 50 years old | 8 (15) | 3 (14) | 5 (17) | 5 (14) | |

1—The relationship between the group of GAG and nGAG respondents and sociodemographic data was analyzed through $X^2$. The analysis was designed to assess the homogeneity of the group.

Over half of the respondents were aged 30 through 39 (52%). The most common place of residence was a city of >100,000 residents (44%). The majority (56%) of the respondents declared a university education. Many women did not use plant galactagogues (58%). There was no statistically significant relationship between the use of galactagogues and respondents' education level, place of residence, or age (Table 1).

## 2.2 Questionnaire

Before final distribution, the questionnaire was pre-tested on 10 test respondents. Following recommendations, a questionnaire was pre-tested to minimize errors from respondents who may misunderstand a question or cannot respond accurately [14]. At that stage, spelling errors, questions that were too long, too complicated, or too demanding, any logical inconsistencies, the use of questions hidden within other questions (double questions), and inaccurate or incomplete skip routines were identified and corrected.

In the initial pre-questionnaire information, the meaning and terminology of the word "galactagogues" were explained. The questionnaire included direct contact with the researchers in case of additional questions or concerns. The respondents were informed that the questionnaire would take about 10 minutes to complete and that the results obtained from the answers would be used exclusively for research purposes.

The questions were closed (11) and open (5). Depending on the closed/open form of the question, information was provided before each question, such as whether the question allows for more than one answer or is a single choice. A note was also included that "if a given question is deemed too personal, the respondent can skip it". The questionnaire was divided into two parts—informational and research. Sociodemographic information and data on lactation use of galactagogues were collected. The questions included in the questionnaire were related

to awareness, prevalence of use, place of purchase, experience with galactagogues, evaluation of effectiveness, incidence of possible side effects, sources of knowledge, and form of use.

## 2.3 Statistical analysis

Statistical analysis was performed using Statistica v. 13.3 statistical software (StatSoft Polska Sp. z o.o. 2023; Kit Plus version 5.0.96; www.statsoft.pl). After considering the confidence level (90%) and the margin of error (14%), the calculated minimum sample size was 35 subjects. The data were categorized. The $X^2$ test (to assess the distribution of categorical variables such as age, education, place of residence, or use and knowledge of galactagogues) and the Spearman test for non-parametric data were used for statistical analysis.

## 3. Results

Among the respondents, almost 70% had heard of galactagogues (Table 2). The most frequently indicated galactagogues (61%) were fennel and aniseed (33%). The source of information about galactagogues for almost half of the respondents who heard about galactagogues was the Internet (42%), whereas information obtained from family (36%) or friends (33%) were also frequent. The overwhelming majority of those using galactagogues (GAGs) took them as ready-made lactation teas (73%). Respondents most often obtained galactagogues from the pharmacy (64%). Herbal or health food stores (36%) or supermarkets/grocery/vegetable stores (27%) were other commonly indicated options. Almost half of the GAG group used galactagogues after recommendations from family and friends (45%). Information from the Internet frequently encouraged the respondents to use plant-based galactagogues (32%). Most of the GAG group confirmed that these remedies effectively stimulated lactation (82%) and did not notice side effects (91%). Non-users of galactagogues (nGAG) did not think they needed to use them. Most respondents in the nGAG group confirmed no need for their use (60%). Respondents often pointed to a lack of knowledge regarding plant galactagogues (50%) as an explanation for not using them. Respondents in the nGAG group said that more knowledge about galactagogues (57%) or lactation problems (37%) would have encouraged them to use galactagogues.

Several relations were detected between factors prompting the GAG respondents to use plant-based galactagogues (Table 3). There was a relation between how galactagogues were recommended (e.g., by family and friends, by information obtained from the Internet, by a doctor) and the age of the respondents $X^2$ (2, N = 22) = 7,9, p = .019, education $X^2$ (3, N = 22) = 8.4, p = .038, and place of residence $X^2$ (3, N = 22) = 8.5, p = .037, respectively. Respondents aged 30–39, characterized by higher education and living in a city of >100,000, were more likely to reach for galactogogues through recommendations from family and friends. Respondents' age was significantly related to knowledge of particular galactagogues $X^2$ (2, N = 35) = 8.5, p = .015. Respondents aged 30–39 were likelier than others to point to fennel. The method of obtaining information on galactagogues (Internet, family, friends, and others) depended on the respondents' place of residence (rural, city <20,000, city 20–100,000, city >100,000) $X^2$ (3, N = 35) = 8.3, p = .040. Respondents residing in cities >100,000 were more likely to rely on information obtained via the Internet from family and friends than the rest of the group.

Based on the results, we found a relation between the place of residence $X^2$ (3, N = 35) = 8.3, p = .040 and the source of knowledge about plant galactagogues. The most frequently chosen option was the Internet. However, it should be remembered that access to the Internet may have been difficult for some respondents. Perhaps they chose to obtain their information from another source in that case. The study also detected a relation between the factor prompting female respondents to use plant galactagogues and age $X^2$ (2, N = 22) = 7,9, p = .019,

**Table 2. Questionnaire results.**

| | Number of total participants N = 52 (%N) | | | |
|---|---|---|---|---|
| | The total study sample n = 52 (100%) | Users (GAG) n = 22 (42%) | Non-users (nGAG) n = 30 (58%) | Women with knowledge of galactagogues (fGAG) n = 36 (69%) |
| Have you heard of plant-derived substances that stimulate lactation (galactagogues)? [1] | | | | |
| Yes | 36 (69%) | 22 (100%) | 14 (47%) | 36 (100%) |
| No | 16 (31%) | 0 (0%) | 16 (53%) | 0 (0%) |
| 1. What plant galactagogues have you heard of? [2] | | | | |
| Fennel | 22 (42%) | 15 (68%) | 7 (23%) | 22 (61%) |
| Aniseed | 12 (23%) | 10 (45%) | 2 (7%) | 12 (33%) |
| Common nettle | 5 (10%) | 2 (9%) | 3 (10%) | 5 (14%) |
| Lactation tea | 5 (10%) | 2 (9%) | 3 (10%) | 5 (14%) |
| Barley malt | 4 (8%) | 4 (18%) | 0 (0%) | 4 (11%) |
| Fenugreek | 3 (6%) | 1 (4%) | 2 (7%) | 3 (8%) |
| Wheat malt | 1 (2%) | 0 (0%) | 1 (3%) | 1 (3%) |
| Cumin | 1 (2%) | 1 (4%) | 0 (0%) | 1 (3%) |
| Basil | 1 (2%) | 0 (0%) | 1 (3%) | 1 (3%) |
| Verbena | 1 (2%) | 0 (0%) | 1 (3%) | 1 (3%) |
| Coffee | 1 (2%) | 0 (0%) | 1 (3%) | 1 (3%) |
| Thistle | 1 (2%) | 1 (4%) | 0 (0%) | 1 (3%) |
| 2. What was the source of your knowledge about galactagogues? [1] | | | | |
| Internet | 15 (29%) | 11 (50%) | 4 (13%) | 15 (42%) |
| Family | 13 (25%) | 7 (32%) | 6 (20%) | 13 (36%) |
| Friends | 12 (23%) | 4 (18%) | 8 (27%) | 12 (33%) |
| Women's magazines | 7 (13%) | 6 (27%) | 1 (3%) | 7 (19%) |
| Physician | 5 (10%) | 3 (14%) | 2 (7%) | 5 (14%) |
| Pharmacist | 5 (10%) | 5 (23%) | 0 (0%) | 5 (14%) |
| Lactation consultant | 5 (10%) | 2 (9%) | 3 (10%) | 5 (14%) |
| Midwife | 1 (2%) | 0 (0%) | 1 (3%) | 1 (3%) |
| Nurse | 1 (2%) | 0 (0%) | 1 (3%) | 1 (3%) |
| 3. If you used galactagogues, do you remember which ones? [2] | | | | |
| Lactation tea | 16 (31%) | 16 (73%) | - | 16 (44%) |
| Fennel | 5 (10%) | 5 (23%) | - | 5 (14%) |
| Barley malt | 2 (4%) | 2 (9%) | - | 2 (6%) |
| Coffee | 2 (4%) | 2 (9%) | - | 2 (6%) |
| I don't remember | 1 (2%) | 1 (5%) | - | 1 (3%) |
| 4. In what form did you (would you) most often use plant-based galactagogues? [1] | | | | |
| Ready-made commercially available teas/plant milk products | 20 (38%) | 20 (91%) | - | 20 (56%) |
| Herbal infusions, prepared by yourself from purchased herbs | 4 (8%) | 4 (18%) | - | 4 (11%) |
| Herbs in loose form | 2 (4%) | 2 (9%) | - | 2 (6%) |
| 5. Where did you most often source galactagogues from? [1] | | | | |
| Pharmacy | 14 (27%) | 14 (64%) | - | 14 (39%) |
| Herbal or health food stores | 8 (15%) | 8 (36%) | - | 8 (22%) |
| Supermarkets/grocery/vegetable stores | 6 (12%) | 6 (27%) | - | 6 (17%) |
| Via the Internet | 3 (6%) | 3 (14%) | - | 3 (8%) |
| Family | 1 (2%) | 1 (4%) | - | 1 (3%) |
| 6. Who or what prompted you to use galactagogues? [1] | | | | |
| Recommendations from family and friends | 10 (19%) | 10 (45%) | - | 10 (28%) |

*(Continued)*

**Table 2.** (Continued)

| | Number of total participants N = 52 (%N) | | | |
| | The total study sample n = 52 (100%) | Users (GAG) n = 22 (42%) | Non-users (nGAG) n = 30 (58%) | Women with knowledge of galactagogues (fGAG) n = 36 (69%) |
|---|---|---|---|---|
| Information obtained from the Internet | 7 (13%) | 7 (32%) | - | 7 (19%) |
| Doctor | 4 (8%) | 4 (18%) | - | 4 (11%) |
| Information obtained from women's magazines | 4 (8%) | 4 (18%) | - | 4 (11%) |
| Lactation consultant | 2 (4%) | 2 (9%) | - | 2 (6%) |
| Advertisements | 1 (2%) | 1 (5%) | - | 1 (3%) |
| Lack of milk | 1 (2%) | 1 (5%) | - | 1 (3%) |
| 7. How do you evaluate the effectiveness of the galactagogues you used? [1] | | | | |
| The remedy stimulated lactation | 18 (35%) | 18 (82%) | - | 18 (50%) |
| Did not stimulate, did not affect lactation | 4 (8%) | 4 (18%) | - | 4 (11%) |
| 8. Have you noticed any "side effects" of using galactagogues?* [2] | | | | |
| No | 20 (38%) | 20 (91%) | - | 20 (56%) |
| Yes | 2 (4%) | 2 (9%) | - | 2 (6%) |
| 9. How do you rate the taste quality of galactagogues? [1] | | | | |
| Tasty | 10 (19%) | 10 (45%) | - | 10 (28%) |
| Their taste did not matter to me at all | 9 (17%) | 9 (41%) | - | 9 (25%) |
| Not tasty | 3 (6%) | 3 (14%) | - | 3 (8%) |
| 10. What was the reason for not using galactagogues? (For women who did not express a desire to use them) [1] | | | | |
| There was no such need | 18 (35%) | - | 18 (60%) | 14 (39%) |
| Lack of knowledge about them | 15 (29%) | - | 15 (50%) | 1 (3%) |
| 11. What would be able to encourage you to use galactagogues? (For women who have not expressed a desire to use them) [2] | | | | |
| Better knowledge about them | 17 (33%) | - | 17 (57%) | 5 (14%) |
| Lactation problems | 11 (21%) | - | 11 (37%) | 8 (22%) |
| Nothing can encourage me | 1 (2%) | - | 1 (3%) | 0 (0%) |
| Examples proving effectiveness | 1 (2%) | - | 1 (3%) | 1 (3%) |

*2 women reported a side effect of diarrhoea affecting their children.

[1] –question with the choice

[2] –open questions

education $X^2$ (3, N = 22) = 8.4, p = .038, and place of residence $X^2$ (3, N = 22) = 8.5, p = .037. The data revealed that women of various ages and places of residence are susceptible to socio-demographic factors. Notably, information obtained through the Internet was the second most frequently chosen factor to encourage the use of plant galactagogues. Once again, the crucial role of the Internet can be seen. In addition, depending on the age of the respondents, differences were seen in the knowledge of individual galactagogues $X^2$ (2, N = 35) = 8.5, p = .015. Respondents often indicated fennel and aniseed as galactagogues with which they were familiar. However, in practice, they chose different products depending on their age. Women aged 30 through 39 pointed to nettle and barley malt, whereas the younger respondents (20 through 29) pointed to lactation mixtures in teas.

## 4. Discussion

Our study examined the knowledge and use of of plant-based substances that stimulate lactation. Lactation represents a period that poses substantial physiological and metabolic challenges for the female body. Notably, lactation requires a great deal of awareness, a balanced

**Table 3. Results of the survey—relations in the group of galactagogues users ($X^2$ (degress of freedom, $N$ = sample size) = chi-square statistic value, $p$ = $p$-value).**

| | Age[7] | Education level[8] | Place of residence[9] |
|---|---|---|---|
| **What plant galactagogues have you heard of?** | | | |
| (1) Fennel | $X^2$ (2, N = 35) | $X^2$ (3, N = 35) | $X^2$ (3, N = 35) |
| (2) Others[1] | = 8.5, p = .015 | = 2.3, p = .517 | = 1.5, p = .694 |
| **What was the source of your knowledge about galactagogues?** | | | |
| (1) Internet, Family | $X^2$ (2, N = 35) | $X^2$ (3, N = 35) | $X^2$ (3, N = 35) |
| (2) Others[2] | = 3.4, p = .180 | = 2.0, p = .582 | = 8.3, p = .040 |
| **If you used herbal remedies to increase lactation, do you remember what they were?** | | | |
| (1) Lactation herbal mixtures in the form of tea | $X^2$ (4, N = 22) | $X^2$ (6, N = 22) | $X^2$ (6, N = 22) |
| (2) Fennel | = 1.8, p = .773 | = 2.8, p = .839 | = 2.7, p = .841 |
| (3) Others[3] | | | |
| **In what form did you most often use plant-based galactagogues?** | | | |
| (1) Ready-made commercially available teas/plant milk products | $X^2$ (2, N = 22) | $X^2$ (3, N = 22) = | $X^2$ (3, N = 22) |
| (1) Others[4] | = .35, p = .838 | 1.1, p = .781 | = 1.8, p = .625 |
| **Where did you most often source galactagogues from?** | | | |
| (1) From pharmacies, herbal or health food stores | $X^2$ (2, N = 22) | $X^2$ (3, N = 22) = | $X^2$ (3, N = 22) |
| (1) Others[5] | = .76, p = .685 | 6.3, p = .097 | = 6.8, p = .078 |
| **Who or what prompted you to use galactagogues?** | | | |
| (1) Recommendations from family and friends | $X^2$ (2, N = 22) | $X^2$ (3, N = 22) = | $X^2$ (3, N = 22) |
| (2) Others[6] | = 7.9, p = .019 | 8.4, p = .038 | = 8.5, p = .037 |

[1]Others: Anise, Stinging nettle, Lactation teas, Barley malt, Fenugreek, Wheat malt, Roman cumin, Basil, Verbena, Coffee, Thistle

[2]Others: Friends, Women's magazines, Doctor, Pharmacist, Lactation consultants, Midwife, Nurses.

[3]Others: Barley malt, Coffee, "I do not remember

." [4]Others: Herbal infusions, self-prepared from bought herbs

[5]Others: Supermarkets/grocery/vegetable stores, Over the Internet, From family

[6]Others: Information obtained from the Internet, Doctor, Information obtained from women's magazines, Lactation consultant, Advertisements, Lack of milk

[7]3 categories—20–29 years old, 30–39 years old, 40–50 years old

[8]4 categories—primary, vocational, secondary, higher education

[9]4 categories—rural, city<20 thousand, city 20–100 thousand, city >100 thousand.

diet, and lifestyle modifications, as does the pregnancy period or trying to get pregnant [15]. In addition, its maintenance during the feeding period requires attention to the amount of fluids and is very often supported by plant lactogenic agents–galactagogues. The production of milk is controlled by the hormone prolactin [16]. Nipple stimulation controls prolactin release, whereas oxytocin controls milk release, which is experienced as a letdown [16]. Human milk production is a complex physiological process that involves physical and emotional factors and the interaction of multiple hormones, the most important of which is believed to be prolactin [17]. Once lactation is established, there is no direct correlation between serum prolactin levels and the volume of milk produced in lactating women [17]. However, the majority of lactating women have a higher baseline prolactin level than non-lactating women for several months and continue to experience suckling-induced peaks when breastfeeding [17]. Galactagogues should be considered for patients with untreatable causes of reduced breast milk production [18]. Many such products have been evaluated based on their mechanism of action, transfer properties to maternal milk, effectiveness, and potential side effects for mother and infant [18]. The perceived effectiveness varied greatly across galactagogues [9]. Perceived effectiveness was highest for domperidone, but more than 23% of domperidone users reported experiencing multiple side effects, compared to an average of 3% of women taking herbal galactagogues [9].

The potential side effects of plant-based galactagogues may include digestive issues, headaches, or dizziness [19]. Fenugreek can also cause severe allergic reactions in some people -particularly those with nut or legume sensitivities- and large doses could cause a dangerous drop in blood sugar levels [19].

According to our survey, most participants were aware of galactagogues. About a third stated that they did not require such stimulants, indicating that they did not experience any lactation issues. However, almost half of the respondents reported using galactagogues. Respondents who used them overwhelmingly found them effective (82%). It is not possible to determine whether the respondents used the galactagogues preventively or due to the occurrence of a lactation disorder. If they did so out of necessity, through an occurring difficulty, the survey results revealed a major common problem with lactation. Considering that in 2020, more than 330,000 children were born in Poland [20], the authors are left to surmise how large a percentage of young mothers struggled with lactation disorders. Our study shed light on the prevalence of lactation issues among breastfeeding women in Poland, emphasizing the need for targeted lactation support. Given that almost half of the respondents reported using galactagogues, it indicated a notable demand for lactation support. That underscored the importance of addressing lactation problems as a public health issue, likely in Poland and other countries.

The use of ready-made herbal lactogenic mixtures in tea was by far the most popular among the respondents. This result was consistent with literature reports, which confirmed that of all lactogenic agents, tea blends were by far the most popular [8]. Familiarity with galactagogues and opinions of their efficacy was consistent in this study, as well as the data found in the literature [8]. Although ready-made tea blends proved to be the most popular, it is worth noting that our study also revealed other ways to stimulate lactation. The respondents also used herbal infusions and powdered herbs, such as fennel and aniseed, as yoghurt toppings. However, such a formulation was far less popular. The preference for herbal lactation support, especially in the form of tea blends, reflects cultural and societal norms related to motherhood and healthcare-seeking behaviors in Poland. Understanding these cultural factors is crucial for tailoring interventions and educational programs to effectively promote breastfeeding and lactation support. The decision to use galactagogues is influenced predominantly by three core and interrelated domains: access to and quality of breastfeeding support, maternal agency, and determination to provide breast milk [21]. Women revealed many problematic experiences with healthcare professionals that left them feeling dismissed and confused due to the provision of inconsistent and insufficient information that was sometimes at odds with their desire to provide breast milk [21]. Some women turned to galactagogues to regain agency [21]. A range of broader influences on the galactagogues' decision-making also emerged. These were separated into categories that related to breast milk supply, which included maternal emotional well-being, social norms and pressures, concerns about infant development, maternal physical health and lactation history, as well as those related specifically to galactagogue use, including: desire for a guaranteed/urgent response, risk-risk trade-off, acceptance and trust, and accessibility and cost [21].

Our data strengthened the importance of the Internet as a source of information for the respondents, which correlated with data in the literature [5]. In a survey conducted in Australia, as many as 35% of respondents identified social media as a popular source of information on plant-based galactagogues [22]. Thus, it is worth considering spreading information about plant-based lactation aids through social media. In our study, the Internet and the recommendations for family and friends were, for many women, factors that effectively encouraged the use of plant-based lactogenic agents. Our results were very much in line with data from Australia, where, similar to the proportions of respondents in our study, Australian women

declared media to be a factor effectively encouraging the use of galactagogues [9]. The study underscored the influential role of the Internet and social media in disseminating information about lactation support. Leveraging such platforms for educational campaigns and awareness programs can reach a wider audience of breastfeeding women in Poland and other regions.

The data obtained among women who do not use galactagogues (nGAGs) also provided important insights. More than half of this group declared that their knowledge (57%) would be essential to encourage using plant-based galactagogues effectively. Therefore, its promotion among breastfeeding mothers appears to be essential for supporting the lactation process. Furthermore, our findings may inform policy decisions related to maternal and child health, including integrating lactation support services into existing healthcare systems, subsidizing lactation products, and implementing breastfeeding-friendly policies in workplaces and public spaces. Policymakers can use these insights to develop comprehensive strategies to promote breastfeeding and support lactating mothers. They may compare the findings of this study with similar research conducted in other European countries or globally, which can provide valuable insights into cross-cultural variations in lactation practices, healthcare systems, and social determinants of health. Such comparative analysis can inform best practices and policy recommendations internationally, contributing to global efforts to improve maternal and child health outcomes.

Notably, Poland has an institution of lactation consultants. Those are usually midwives, nurses, or doctors who have expanded their lactation knowledge postgradually [23]. Consultants can help breastfeeding women through appropriate diagnosis and therapy in case of lactation disorders and problems. Lactation consultants are certified as CDL (certified lactation consultant)—awarded to trained experts by the Center for Lactation Science [23]. Lactation consultants should distribute knowledge about plant-based galactagogues. Unfortunately, this option was pointed out only five times, representing 14% of the respondents who had heard about galactagogues from lactation consultants. Compared to sources such as the Internet (42%), family (36%), or friends (33%), this result is alarmingly low. Rather than to a lactation consultant, respondents even pointed to women's magazines as their source of information. This result raises thoughts about the availability and knowledge of lactation consultants in hospital gynaecology and obstetrics departments. The low utilization of lactation consultants among respondents highlights a potential gap in healthcare services. Healthcare providers, including midwives, nurses, and doctors, must know about lactation support, including plant-based galactagogues, to assist breastfeeding mothers better. This may entail incorporating lactation education into medical training programs and encouraging ongoing professional development for healthcare practitioners.

In the questionnaire, when asked about the source of knowledge about plant galactagogues to choose from, the option "nutritionist" was also given. Not once was it selected. Perhaps the respondents did not take nutritional advice from a dietitian during their pregnancy, or the dietitians they worked with did not know about galactagogues and could not offer such options to the patients.

The study results indicate the need to popularize plant galactagogues among women. The survey was conducted during the SARS-CoV-2 pandemic when the sanitary-epidemiological restrictions and strictures markedly limited the possibility of conducting surveys traditionally —through direct contact with respondents. In order to reduce the likelihood of infection and safeguard the well-being of both interviewers and survey participants, it was essential to survey in an online format. This approach allowed for data collection without jeopardizing the health or privacy of the individuals involved. At the same time, respondents could complete the survey at any time and place that was safe for them.

Nevertheless, it should be stressed that, due to the distribution mode, it could only be accessed by women who assessed social media platforms, forums, discussion groups, or emails. This factor may have had an impact on the size of the sample that was collected. Although it was only possible to survey 52 respondents in the study, the representativeness of this group may be questionable due to access to the Internet. Nevertheless, it should be noted that the sample was collected during the pandemic when access to hospitals and contact with women in gynaecological and obstetric wards was difficult. In addition, the study sample was diverse in terms of age, education, and place of residence. This provided an opportunity to assess the relationships among the analyzed variables. Our study provides valuable insights into the knowledge and use of plant-based galactagogues among Polish women, but further research is needed to explore the effectiveness and safety of these interventions. Longitudinal studies to assess the impact of different galactagogues on lactation outcomes and qualitative research to understand women's experiences and preferences can inform evidence-based recommendations for lactation support.

In conclusion, plant-based galactagogues in the form of herbal lactation mixtures in tea, as the most popular in the group of breastfeeding mothers, effectively boosted the lactation of respondents with minimal declared side effects. However, spreading the knowledge about galactagogues is also essential, as some respondents declared that more knowledge about galactagogues would be an influential factor in encouraging their use.

## 5. Conclusions

Lactating women were predominantly positive about the use of galactagogues. A slightly smaller group of these women used galactagogues. Almost all women who used them confirmed their effectiveness in stimulating lactation. The results confirm the need to educate young mothers using mass media tailored to their age and location. Educating nutritionists about plant-based galactagogues for patients struggling with lactation problems is also essential. The study's insights extend beyond Poland and have implications for healthcare practice, policy, and research in Europe and beyond; it highlights the importance of addressing lactation issues and promoting breastfeeding as a public health priority.

## Author Contributions

**Conceptualization:** Agnieszka Garbacz, Paweł Juszczak, Magdalena Człapka-Matyasik.

**Data curation:** Agnieszka Garbacz, Magdalena Człapka-Matyasik.

**Formal analysis:** Agnieszka Garbacz.

**Funding acquisition:** Marcin Nowicki, Przemysław Łukasz Kowalczewski.

**Investigation:** Agnieszka Garbacz.

**Methodology:** Agnieszka Garbacz.

**Resources:** Marcin Nowicki, Przemysław Łukasz Kowalczewski.

**Software:** Agnieszka Garbacz.

**Supervision:** Paweł Juszczak, Magdalena Człapka-Matyasik.

**Visualization:** Agnieszka Garbacz.

**Writing – original draft:** Agnieszka Garbacz.

**Writing – review & editing:** Agnieszka Garbacz, Paweł Juszczak, Marcin Nowicki, Przemysław Łukasz Kowalczewski, Magdalena Człapka-Matyasik.

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
