## [Decision Letter · Decision Letter 0]

16 Jul 2024

PONE-D-24-05417Exploring Galactagogue Use Among Breastfeeding Women: Insights from a Cross-Sectional StudyPLOS ONE

Dear Dr. Nowicki,

Thank you for submitting your manuscript to PLOS ONE. After careful consideration, we feel that it has merit but does not fully meet PLOS ONE’s publication criteria as it currently stands. Therefore, we invite you to submit a revised version of the manuscript that addresses the points raised during the review process.

We look forward to receiving your revised manuscript.

Kind regards,

Solomon Tesfay

Academic Editor

PLOS ONE

3. In the online submission form you indicate that your data is not available for proprietary reasons and have provided a contact point for accessing this data. Please note that your current contact point is a co-author on this manuscript. According to our Data Policy, the contact point must not be an author on the manuscript and must be an institutional contact, ideally not an individual. Please revise your data statement to a non-author institutional point of contact, such as a data access or ethics committee, and send this to us via return email. Please also include contact information for the third party organization, and please include the full citation of where the data can be found.

Reviewers' comments:

Reviewer's Responses to Questions

**Comments to the Author**

1. Is the manuscript technically sound, and do the data support the conclusions?

Reviewer #1: Partly

Reviewer #2: Yes

2. Has the statistical analysis been performed appropriately and rigorously? 

Reviewer #1: Yes

Reviewer #2: Yes

3. Have the authors made all data underlying the findings in their manuscript fully available?

Reviewer #1: Yes

Reviewer #2: Yes

4. Is the manuscript presented in an intelligible fashion and written in standard English?

Reviewer #1: Yes

Reviewer #2: Yes

5. Review Comments to the Author

Reviewer #1: Thank you for taking the time to consider me as a potential reviewer for your manuscript. After carefully analyzing the content, I believe that my input could significantly strengthen the manuscript. I have identified several areas that could benefit from further clarification and expansion, and I am confident that my comments could help enhance the overall quality of the work.

The exact questions or the nature of the questions should be outlined. If the questionnaire included both closed and open questions or was divided into sections, this should be described in details.

Any instructions that have been given on how to complete the questionnaire should be provided to ensure consistency in how participants respond.

Any steps taken to validate or test the questionnaire before its final distribution should be mentioned.

Specifics on how the online distribution was carried out, such as through specific forums, social media platforms, or emails, can help understand the reach and potential biases in data collection.

Information on response rates, any reminders sent to participants, or strategies used to maximize response rates should be described.

How data confidentiality was ensured beyond the anonymity stated.

The rationale behind choosing an online distribution method and Google Forms.

Reviewer #2: Thank you for the opportunity to review this paper.

The study presented is the results of an online survey in which 52 women participated. Unfortunately, it is not a cross-sectional study, as the title suggests.

In a cross-sectional study, the study would measure the outcome and the exposures in the participants at the same time - this does not appear to have been what occurred in this study - as participants either were breastfeeding at the time of the study or had breastfed in the past (and I assume taken or not taken galactagogues at these other points as well). It is also unclear what population was studied as no details were provided on how the survey was distributed aside from "online" - I assume participants were from Poland?

In its current form, with limited details about the study design (and erroneously labelled as a cross-sectional study) and responses from only 52 women, it is not possible, in my opinion, to recommend the paper for publication.

6. PLOS authors have the option to publish the peer review history of their article (what does this mean?). If published, this will include your full peer review and any attached files.

Reviewer #1: **Yes: **Ziyad Saeed Almalki

Reviewer #2: No

---

## [Author Response · Author response to Decision Letter 0]

14 Aug 2024

Authors' Response to the Reviewers' Comments

Journal: PLOS ONE

Title: Exploring Galactagogue Use Among Breastfeeding Women: Insights from an Observa-tional Study

Authors: Agnieszka Garbacz, Paweł Juszczak, Marcin Nowicki, Przemysław Kowalczewski, and Magdalena Człapka-Matyasik 

Dear PLOS ONE Reviewers and Editors,

We would like to extend our gratitude for your valuable feedback and constructive criticism of our manuscript titled "Exploring Galactagogue Use Among Breastfeeding Women: Insights from an Observational Study". Your time, effort, and expertise in reviewing our work are greatly appreciated.

We have carefully considered the comments and suggestions provided. We are pleased to inform you that we have addressed all the concerns raised and have made appropriate revisions to improve the quality and clarity of the manuscript. Your insightful remarks have contributed to enhancing our report's overall coherence and rigour. We are truly grateful for your thorough examination and thoughtful recommendations, which have strengthened the scholarly integrity of our work.

Please find attached the revised version of our manuscript with detailed responses to each reviewer’s comments. Thank you once again for your time, expertise, and continued support. 

Reviewer 1 – Dr. Ziyad Saeed Almalki

The exact questions or the nature of the questions should be outlined. If the questionnaire included both closed and open questions or was divided into sections, this should be described in details. Information on the type of questions, nature of the questions and division into sections was added and can be found in lines 127-136.

Any instructions that have been given on how to complete the questionnaire should be provided to ensure consistency in how participants respond. Information has been added - line 127-130.

Any steps taken to validate or test the questionnaire before its final distribution should be mentioned. Information has been added - line 114.

Specifics on how the online distribution was carried out, such as through specific forums, social media platforms, or emails, can help understand the reach and potential biases in data collection. Information on how to distribute the questionnaire was added - lines 94-101.

Information on response rates, any reminders sent to participants, or strategies used to maximize response rates should be described. The questionnaire was distributed via the Internet as a link. Due to the lack of a maximum number/quantity of sheets sent out, it is not possible to calculate a response rate. Information concerning reminders was placed in lines 99-101.

How data confidentiality was ensured beyond the anonymity stated. Google Forms numbered completed questionnaires according to order; they were filled by respondents. Despite filling out a questionnaire created in Google Forms, the respondent did not provide email address. The design of questionnaires created in google forms ensures complete anonymity when the option 'do not collect email addresses' is selected). This condition was met and was indicated in the work (line 90-93).

The rationale behind choosing an online distribution method and Google Forms. Justification for the choice of the online distribution method is provided in the revised text - Lines 287-297.

Google Forms allows to create an easy-to-use questionnaire, which is also easy to share. The tool is free, but the created questionnaire can be attractively personalized - for example, by placing questions in different formats. What's more, the data collected through the questionnaire is stored on Google servers (backups possible).

Reviewer 2 - Anonymous

The study presented is the results of an online survey in which 52 women participated. Unfortunately, it is not a cross-sectional study, as the title suggests.

In a cross-sectional study, the study would measure the outcome and the exposures in the participants at the same time - this does not appear to have been what occurred in this study - as participants either were breastfeeding at the time of the study or had breastfed in the past (and I assume taken or not taken galactagogues at these other points as well). Thank you for this comment; we replaced the cross-sectional with observational. 

Nevertheless, we would like to point out that cross-sectional studies include surveys and health assessments conducted to determine the proportion of individuals with certain conditions or behaviours at a given time. The key features of cross-sectional studies, in our feeling we had filled, include:

1. Single Point in Time: Data were collected from participants at one specific time.

2. Prevalence Measurement: They were used to measure the prevalence of intake galactagogues at certain time in life (breastfeeding)

3. Association Identification: They could help identify associations between using/knowledge of galactagogues, background, residence, etc., but they do not establish causality.

4. Observational Nature: We observed and recorded information without manipulating the study environment or subjects.

5. Population Representation: They aimed to provide a snapshot representing a larger population.

It is also unclear what population was studied as no details were provided on how the survey was distributed aside from "online" – I assume participants were from Poland? Information on how to distribute the questionnaire was added (line 94-101). We cannot assume that all respondents were from Poland. The questionnaire was distributed via the Internet using the snowball method. It could be filled out by respondents who no longer live in Poland (the questionnaire was in Polish). All those info was pointed.

---

## [Editor Report · Decision Letter 1]

8 Sep 2024

Exploring Galactagogue Use Among Breastfeeding Women: Insights from an Observational Study

PONE-D-24-05417R1

Dear Dr. Nowicki,

We’re pleased to inform you that your manuscript has been judged scientifically suitable for publication and will be formally accepted for publication once it meets all outstanding technical requirements.

Kind regards,

Solomon Tesfay

Academic Editor

PLOS ONE
---

## [Editor Report · Acceptance letter]

11 Oct 2024

PONE-D-24-05417R1 

PLOS ONE

Dear Dr. Nowicki, 

I'm pleased to inform you that your manuscript has been deemed suitable for publication in PLOS ONE. Congratulations! Your manuscript is now being handed over to our production team.

Kind regards, 

on behalf of

Dr. Solomon Tesfay 

%CORR_ED_EDITOR_ROLE%

PLOS ONE